# Revealing the structure of information flows discriminates similar animal social behaviors

**Gabriele Valentini[1,2], Nobuaki Mizumoto[2,3], Stephen C Pratt[2,4], Theodore P Pavlic[2,4,5,6,7]\*, Sara I Walker[1,4,5]\***

[1]Arizona State University, School of Earth and Space Exploration, Tempe, United States; [2]Arizona State University, School of Life Sciences, Tempe, United States; [3]Okinawa Institute of Science & Technology Graduate University, Onna-son, Okinawa, Japan; [4]Arizona State University, ASU–SFI Center for Biosocial Complex Systems, Tempe, United States; [5]Arizona State University, Beyond Center for Fundamental Concepts in Science, Tempe, United States; [6]Arizona State University, School of Computing, Informatics, and Decision Systems Engineering, Tempe, United States; [7]Arizona State University, School of Sustainability, Tempe, United States

**Abstract** Behavioral correlations stretching over time are an essential but often neglected aspect of interactions among animals. These correlations pose a challenge to current behavioral-analysis methods that lack effective means to analyze complex series of interactions. Here we show that non-invasive information-theoretic tools can be used to reveal communication protocols that guide complex social interactions by measuring simultaneous flows of different types of information between subjects. We demonstrate this approach by showing that the tandem-running behavior of the ant *Temnothorax rugatulus* and that of the termites *Coptotermes formosanus* and *Reticulitermes speratus* are governed by different communication protocols. Our discovery reconciles the diverse ultimate causes of tandem running across these two taxa with their apparently similar signaling mechanisms. We show that bidirectional flow of information is present only in ants and is consistent with the use of acknowledgement signals to regulate the flow of directional information.

**\*For correspondence:**
tpavlic@asu.edu (TPP);
sara.i.walker@asu.edu (SIW)

**Competing interests:** The authors declare that no competing interests exist.

## Introduction

Social interactions among individuals unfold across different scales of space and time (*Flack, 2012*). At short time scales, causal relationships can often be captured by experiments that manipulate an immediate stimulus to reveal its causal connection(s) to a stereotyped response, or fixed action pattern. In herring gulls, for example, the feeding behavior of chicks is visually triggered by a red spot on the lower bill of adult gulls—a causal relationship revealed through experiments in which changes in the color of this spot were shown to affect the likelihood that chicks will engage in feeding behavior (*Tinbergen, 1953*). However, interactions are often more complex than this because they follow a protocol where rules are conditionally applied over time contingent upon the outcome of previous interactions. In these cases, where short-term histories may affect longer-term outcomes, the ability to make testable predictions requires quantitative tools that can capture the dynamics of the interaction protocol at an intermediate time scale. Considering only short-term interactions, like the stimulus-and-response of herring-gull parent and chick, might not explain functional differences observed at long time scales in otherwise similar behaviors.

**eLife digest** Social animals continuously influence each other's behavior. Most of these interactions simply consist in an individual immediately responding to the behavior of another in a predictable way. Still, when the same individuals interact over long periods, complex social interactions can arise. These can be difficult for scientists to study, because how animals behave at a given moment depends on their shared history.

Certain species of ants and termites use smell and touch to do 'tandem runs' and move in pairs through the environment. Only ants, however, can learn a new route from their running partner. Understanding how this difference arises means examining how the animals interact and communicate over longer time scales. This requires new approaches to capture how information flows between the insects.

Here, Valentini et al. used a scientific methodology known as information theory to study tandem running in one species of ants and two species of termites. Information theory provides a framework to quantify how information is shared, processed and stored.

The flow of information between individuals was measured separately for different aspects of tandem running. At small time scales, ant and termite behavior appeared identical, but over longer periods, it was possible to distinguish between the two types of insects.

In termites, only one individual in a pair sent information to the other to instruct the second termite where to go. By contrast, in ants, both members of the tandem communicated with each other in a way that was consistent with how humans acknowledge information they receive from other individuals.

The approach used by Valentini et al. will be useful to researchers who study how complex and often cryptic social interactions develop over extended periods in social animals. This framework could also be applied in other systems such as groups of cells, or economic networks.

Consider the tandem-running behavior of many ants and termites, in which one individual leads a follower through their environment, the follower walking closely behind the leader throughout the run. (*Figure 1A*). At short time scales, tandem runs in the ant *Temnothorax rugatulus* appear identical to those in the termites *Coptotermes formosanus* and *Reticulitermes speratus*. They use similar signaling mechanisms, in which the leader releases a short-range pheromone that attracts the follower (*Möglich et al., 1974*; *Bordereau and Pasteels, 2010*), while the follower taps the leader's body with its antennae to indicate its continued presence (*Möglich et al., 1974*; *Franks and Richardson, 2006*; *Nutting, 1969*; *Vargo and Husseneder, 2009*). Upon removal of the follower, the leader stops and waits for the follower to resume contact both in ants (*Möglich et al., 1974*; *Franks and Richardson, 2006*) and in termites (*Mizumoto and Dobata, 2019*). At long time scales, however, there exist clear functional differences in these seemingly similar behaviors. Leaders in *T. rugatulus* use tandem runs as a recruitment mechanism that allows followers to learn a route and acquire navigational information necessary to later repeat the same journey independently of the leader (*Figure 1B*). In contrast, termites usually use tandem runs during mating, except for one example of recruitment in a basal termite (*Sillam-Dussès et al., 2007*). In a mated pair, the male follows the female leader only to maintain spatial cohesion when searching for a new home; once a suitable location is found, the termites remain there to start a new colony, and neither partner ever retraces the route of their tandem run.

Short-term signaling mechanisms (*i.e.*, their stimulus and response dynamics) are similar between ants and termites and cannot explain species differences in the function of tandem runs (*i.e.*, route learning versus spatial cohesion). These differences are likely encoded at intermediate time scales, where it becomes possible in principle to detect the communication protocol (*i.e.*, set of interaction rules) that describes how and when leader and follower use each short-term signaling mechanism. However, experimental manipulations that operate at intermediate time scales also interfere with and constrain normal patterns of behavior over time. As we show in this study, information-theoretic methods can reveal the structure of information flow between subjects based only on observational data from many repeated interactions. Moreover, these model-free methods do not rely on a priori

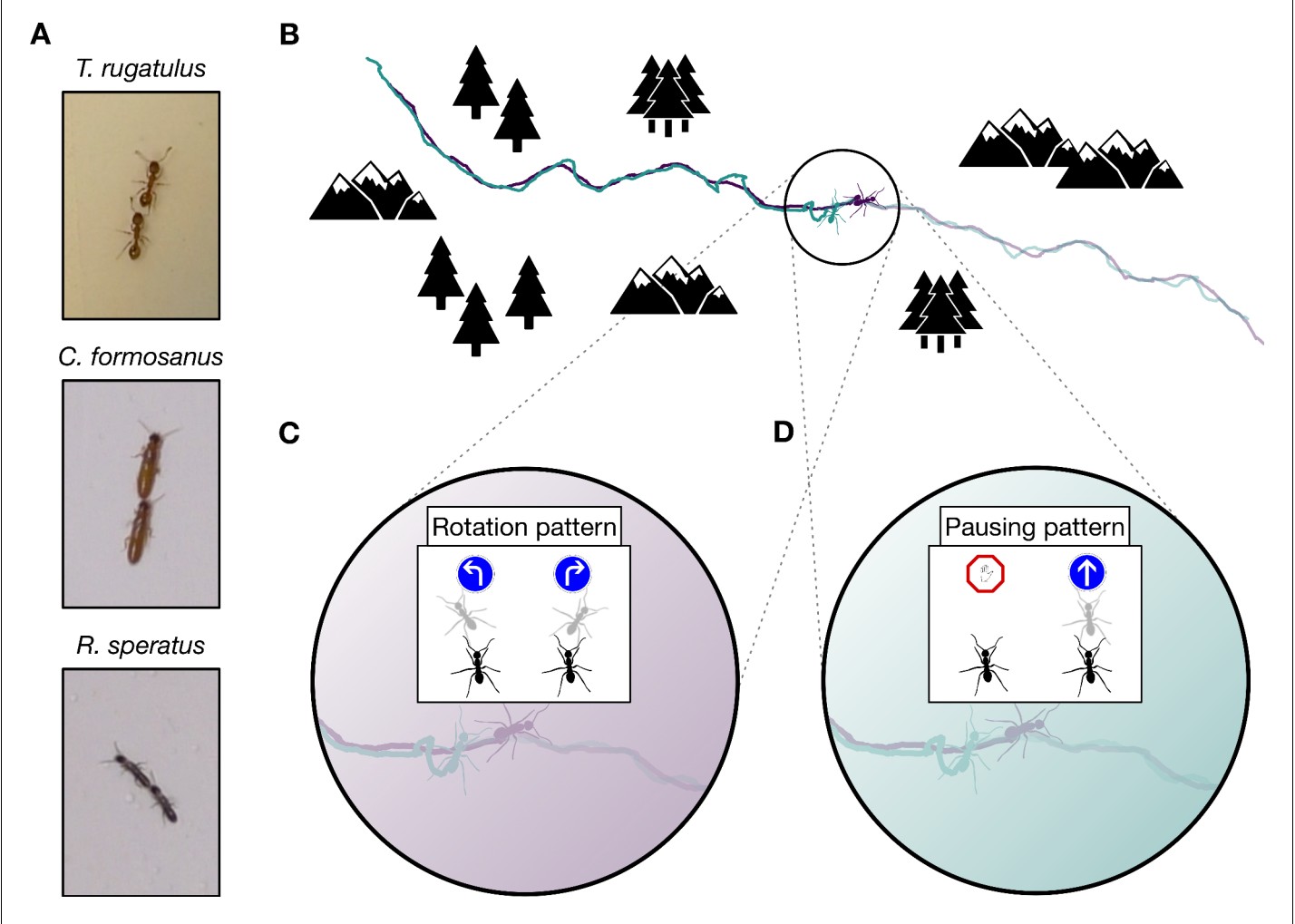

**Figure 1.** Tandem run recruitment by ants and termites. (**A**) Tandem running pairs of the ant species *T. rugatulus* and the termite species *C. formosanus* and *R. speratus*. (**B**) A sampled tandem run trajectory within an idealized environment. The encoding schemas used to discretise spatial trajectories of each ant and each termite on the basis of (**C**) the rotation pattern and (**D**) the pausing pattern.

assumptions and can be applied over different ecological scenarios allowing for comparisons across a wide taxonomic range (*McCowan et al., 1999*).

Information theory provides a model-free formalism to explicitly quantify the effects of the interaction between individuals across space and time (*Cover and Thomas, 2005*; *Lizier et al., 2008*). Whereas the generic concept of *entropy* quantifies the uncertainty in a distribution of outcomes, the derived construct of *transfer entropy* quantifies the reduction of uncertainty about the future state of a putative receiver given knowledge of the present state of the corresponding sender (*Schreiber, 2000*). Transfer entropy is well suited for studying message passing; it naturally incorporates temporal ordering, from the sender's present to the receiver's future, and quantifies the additional predictive power gained from the sender beyond what is contained in the receiver's past. In this way, it accounts for autocorrelations that might otherwise affect behavioral data (*Mitchell et al., 2019*). Previous studies have used symbolic transfer entropy (*Staniek and Lehnertz, 2008*) to reveal whether one animal is influencing another on the basis of a single symbolic representation of behavioral data (*Orange and Abaid, 2015*; *Butail et al., 2016*; *Kim et al., 2018*; *Ward et al., 2018*; *Porfiri et al., 2019*; *Ray et al., 2019*). We extend these methods by applying transfer entropy to different symbolic representations of the same data to capture parallel information flows within the same behavior (*e.g.*, patterns embedded in symbols representing the direction of motion or the speed of motion as shown in *Figure 1C and D*). Using different symbolic representations of the

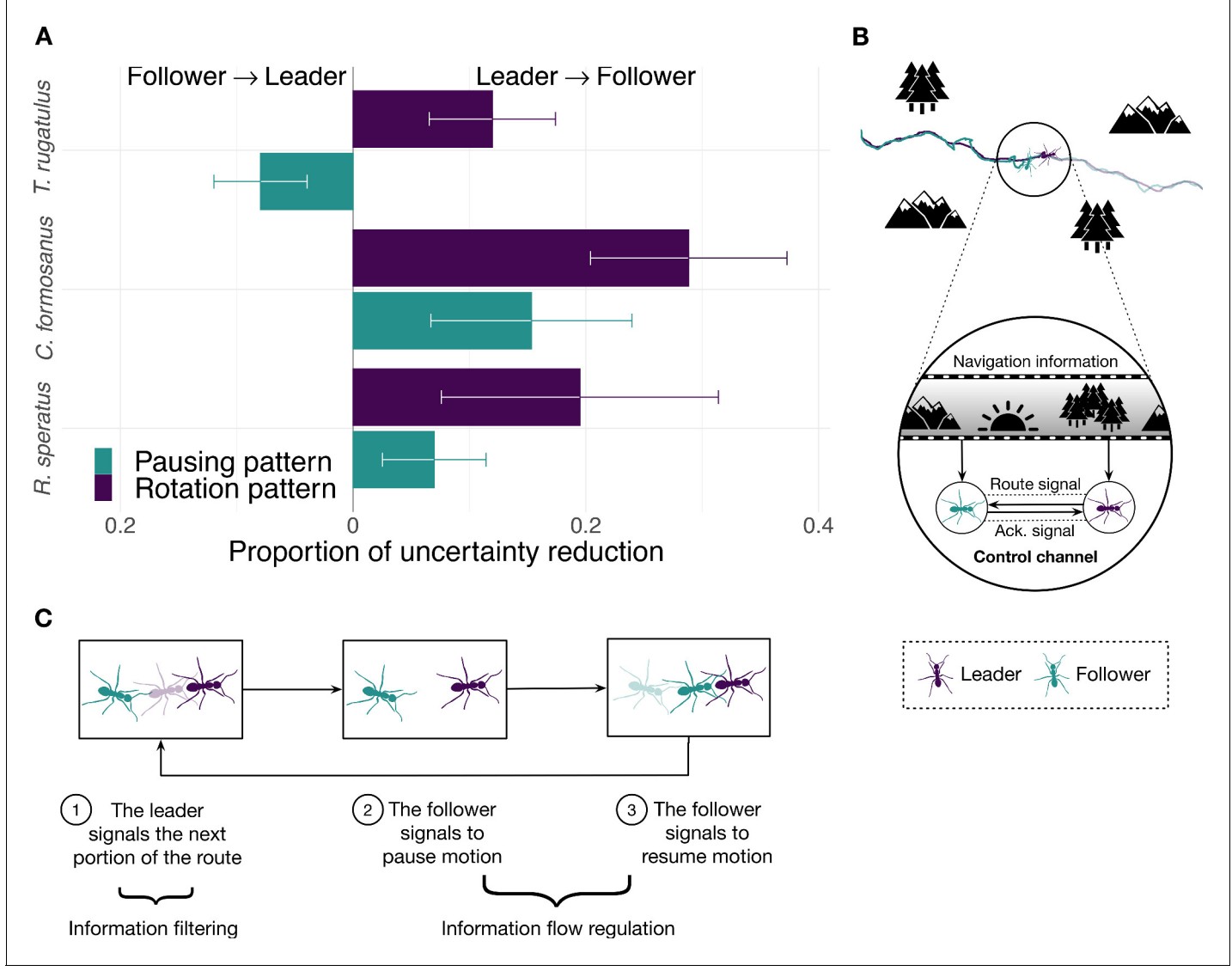

**Figure 2.** Information flow during tandem run recruitment. (**A**) The predominant direction of predictive information given by the proportion of uncertainty reduction explained by the interaction between leader and follower (mean normalized transfer entropy and standard error). (**B**) A schematic illustration and (**C**) a mechanistic illustration of the regulation of information flow in ants' tandem runs. Source data of predictive information are available in *Figure 2—source data 1*.

The online version of this article includes the following source data for figure 2:

**Source data 1.** Conditional entropy and transfer entropy for selected parameter configurations.

same raw data allows us to uncover the complex, multi-layered structure of causal relationships between subjects. Following this approach, we provide evidence that the communication protocol used by leaders and followers over intermediate time scales explains the functional differences between the tandem runs of ants and termites despite their using similar signaling mechanisms at short time scales.

## Results

We first used transfer entropy to find whether the leader's or the follower's behavior better predicts the *direction of motion* of the other runner along the route. In ants, the leader is demonstrating a known route to the follower (*Franks and Richardson, 2006*), and in termites the leader is directing a random search for a new home across the environment with the follower in tow (*Nutting, 1969*;

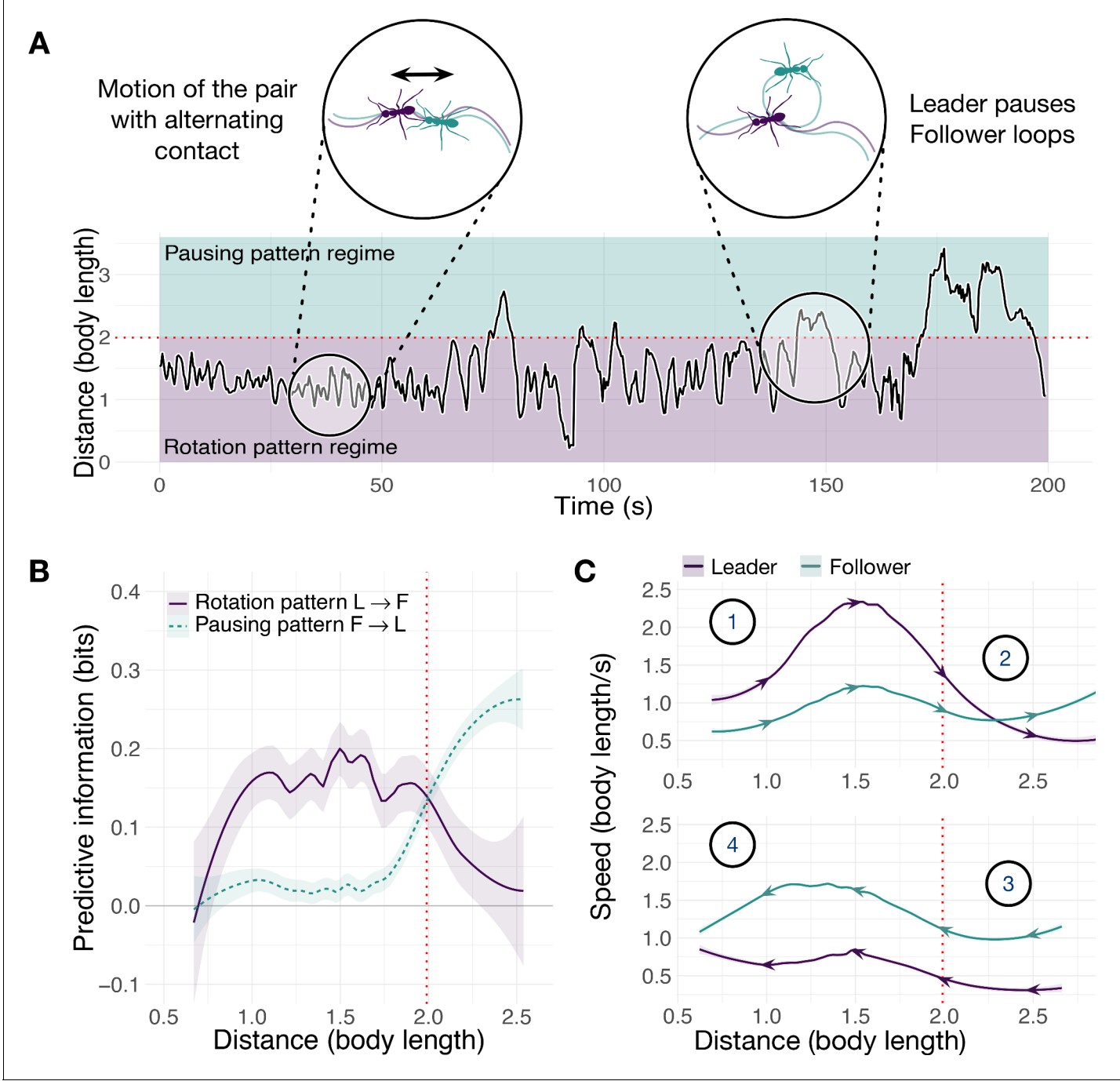

**Figure 3.** Spatiotemporal dynamics of the ant *T. rugatulus*. (**A**) Distance between the centroids of runners as a function of time. (**B**) Average predictive information (measured with local transfer entropy) as a function of the distance between centroids of runners (smoothed conditional means with LOESS smoothing and 0.3 span). (**C**) Average speed of leader and follower as a function of the distance between their centroids for increasing and decreasing distance. Purple represents the leader, green represents the follower. Source data of predictive information are available in *Figure 3—source data 1*. The online version of this article includes the following source data for figure 3:

**Source data 1.** Average local transfer entropy as a function of distance between runners.

*Vargo and Husseneder, 2009*). In both cases, the leader is expected to be the best predictor of the direction of the pair's motion. Consequently, we expect the leader's behavior to be more informative about the direction of the follower than the other way around in both ants and termites. To test this hypothesis, we coarse grained the spatial trajectories of each runner into sequences of clockwise

**Table 1.** Selected parameter configurations.

For each species and behavioral pattern, the table provides the number of tandem runs used in the study, the sampling period and corresponding number of time steps in each time series, and the history length.

| Species | #Tandem runs | Behavioral pattern | Sampling period (s) | #Time steps | History length $k$ |
|---|---|---|---|---|---|
| *T. rugatulus* | 20 | Rotation | 1.5015 | 599 | 9 |
| | | Pausing | 0.9676 | 930 | 13 |
| | | Pausing and Rotation | 1.2346 | 728 | 8 |
| *C. formosanus* | 17 | Rotation | 0.3670 | 2452 | 2 |
| | | Pausing | 0.1668 | 5395 | 1 |
| | | Pausing and Rotation | 0.3670 | 2452 | 2 |
| *R. speratus* | 20 | Rotation | 0.5005 | 1798 | 1 |
| | | Pausing | 0.4671 | 1926 | 1 |
| | | Pausing and Rotation | 0.5005 | 1798 | 2 |

and counterclockwise turns (*Figure 1C* and Materials and methods). We then measured the flow of information between the pair averaged over the entire duration of the tandem run (*i.e.*, over intermediate time scales). We found that, as expected, the leader better predicts the direction of motion of the follower than vice versa across all three species (*Figure 2A*, rotation bars, and *Tables 1* and *2*).

Next, we focused on the frequent brief interruptions that give tandem runs a distinctive stop-and-go appearance. During these interruptions, the follower breaks tactile contact with the leader, who then pauses while the follower performs a local random search (*Franks et al., 2010*; *Mizumoto and Dobata, 2019*). When the follower again touches the leader, the latter resumes motion, and the pair continues on their way. In ants, these frequent interruptions are believed to regulate the speed of the run to better enable followers to acquire navigational information (*Franks and Richardson, 2006*; *Franklin et al., 2011*). As termites do not use tandem runs to learn a route, interruptions may be more consistent with accidental chance separations from the leader. Thus, we hypothesize that in ants, but not in termites, followers better predict the cessation and resumption of motion than do leaders. Under this hypothesis, followers send acknowledgment signals (*Figure 2B and C*) similar to the use of utterances (*e.g.*, 'mm-hmm' described by *Jefferson, 1984*) or gestures (*e.g.*, the nodding of one's head) in human conversations, as well as 'ACK' messages in Internet protocols that confirm receipt of other content-laden packets (*Cerf and Kahn, 1974*). If our hypothesis is correct, we would expect the information-theoretic signature of the tandem pair's pausing pattern in ants to differ from that of termites. To test this, we analyzed the spatial trajectories using a different representation obtained by coarse graining them into sequences of *pauses and movements* (*Figure 1D*). As hypothesized, we found that the leader remains the best source of predictive information in termites, but in ants the follower instead controls the flow of information and better predicts the future pausing behavior of the leader (*Figure 2A*, pausing bars, and *Table 2*).

Side-by-side comparison of tandem-run trajectories (*Figures 3A* and *4A*) shows that ants, but not termites, evince a tension between cohesion and information acquisition. Leader and follower ants repeatedly switch in and out of proximity regulation under the control of the follower (*Figure 3B and C*). The predictive power of the leader's rotation pattern dominates at close distances up to two body lengths, when the pair is undergoing sustained motion and seeking cohesion (point 1, rotation regime). When their distance increases further, the follower becomes more informative, predicting pauses in the motion of the leader (point 2, pausing regime). Their separation then decreases as the follower approaches the stationary leader (point 3) and predicts her resumption of motion. When leader and follower are again in close proximity, the leader begins to move away (point 4) and this pattern repeats. Large separations are evidently generated by the follower ant and are unrelated to rotational course corrections.

In contrast to ants, the termite leader dominates both regimes of predictive information (*Figure 4*). Furthermore, these regimes are inverted with respect to ants: rotation is predicted at larger

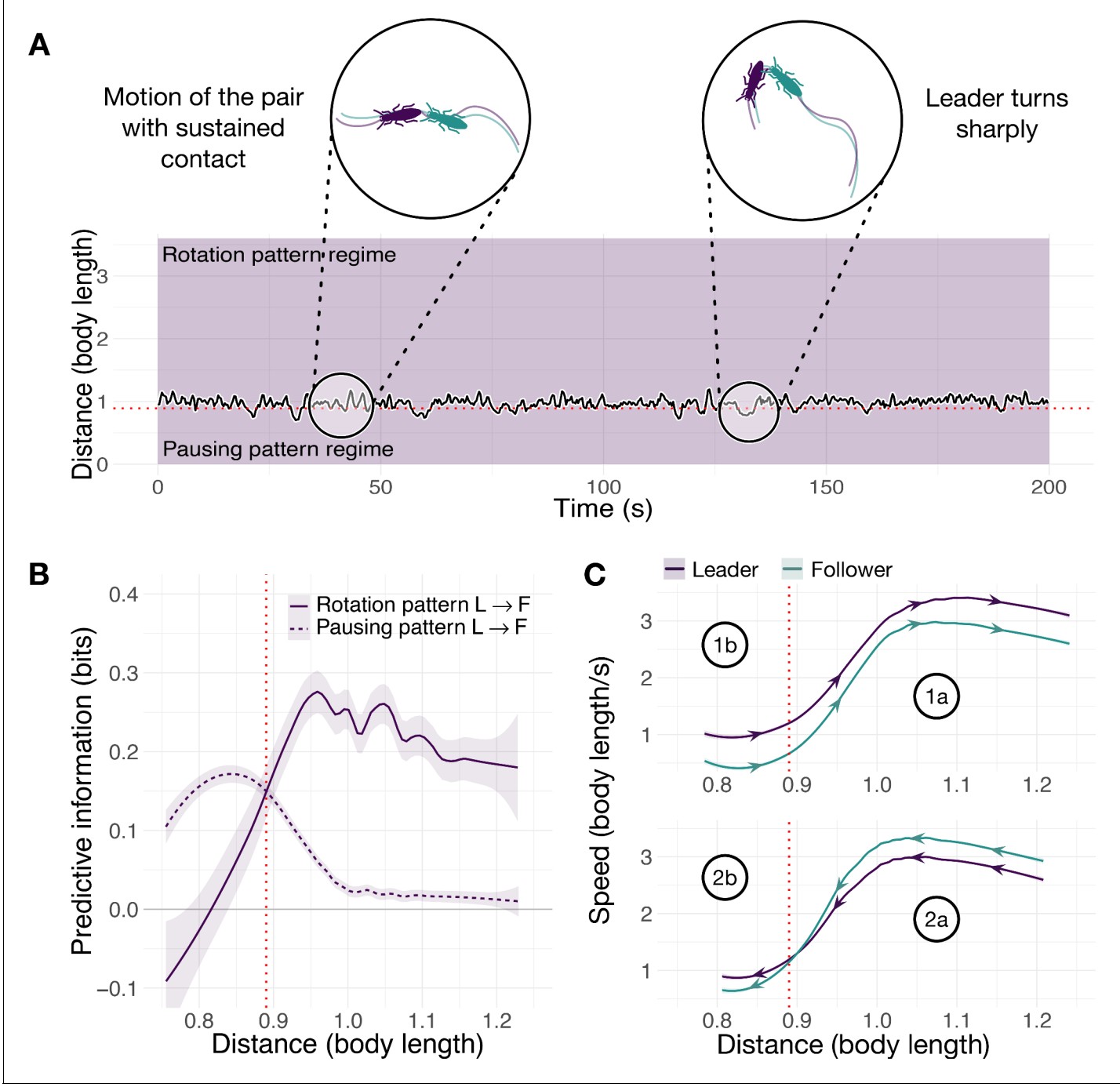

**Figure 4.** Spatiotemporal dynamics of the termite *C. formosanus*. (**A**) Distance between the centroids of runners as a function of time. (**B**) Average predictive information (measured with local transfer entropy) as a function of the distance between centroids of runners (smoothed conditional means with LOESS smoothing and 0.3 span). (**C**) Average speed of leader and follower as a function of the distance between their centroids for increasing and decreasing distance. Purple represents the leader, green represents the follower. See also *Figure 4—figure supplement 1*. Source data of predictive information are available in *Figure 4—source data 1* and *Figure 4—source data 2*.

The online version of this article includes the following source data and figure supplement(s) for figure 4:

**Source data 1.** Average local transfer entropy as a function of distance between runners for *C. formosanus*.
**Source data 2.** Average local transfer entropy as a function of distance between runners for *R. speratus*.
**Figure supplement 1.** Spatiotemporal dynamics of the termite *R. speratus*.

**Table 2.** Statistics about transfer entropy.

Mean value and standard error of transfer entropy for the experimental ($T_{X \to Y}$) and surrogate ($T^s_{X \to Y}$) datasets and of normalized transfer entropy ($\tilde{T}_{X \to Y}$) from the predominant source of predictive information to its destination (*i.e.*, always from leader to follower except for the pausing pattern of *T. rugatulus* when information is transferred from follower to leader).

| Species | Behavioral pattern | $T_{L \to F}$ | $T^s_{L \to F}$ | $T_{F \to L}$ | $T^s_{F \to L}$ | $\tilde{T}_{L \to F}(\tilde{T}_{F \to L})$ |
|---|---|---|---|---|---|---|
| T. rugatulus | Rotation | .1488 ± .0504 | .0366 ± .0026 | .0368 ± .0125 | .0371 ± .0073 | .1197 ± .0543 |
| | Pausing | .0058 ± .0033 | .0047 ± .0015 | .0423 ± .021 | .0081 ± .0044 | .0795 ± .0401 |
| | Pausing and Rotation | .2656 ± .034 | .1429 ± .0092 | .1836 ± .0368 | .1847 ± .0269 | .1246 ± .035 |
| C. formosanus | Rotation | .2063 ± .0731 | .0009 ± .0019 | .0091 ± .0077 | .0012 ± .0017 | .2884 ± .0844 |
| | Pausing | .0303 ± .0209 | .0001 ± .0003 | .0064 ± .0056 | .0001 ± .0001 | .1532 ± .0865 |
| | Pausing and Rotation | .2593 ± .072 | .0024 ± .0028 | .013 ± .0135 | .0026 ± .0014 | .2991 ± .0583 |
| R. speratus | Rotation | .1804 ± .1097 | .0001 ± .0003 | .0027 ± .0137 | .0001 ± .0003 | .1949 ± .119 |
| | Pausing | .0228 ± .0149 | .0001 ± .0002 | .0055 ± .0087 | .0001 ± .0001 | .0698 ± .0445 |
| | Pausing and Rotation | .2385 ± .1446 | .0011 ± .0013 | .0081 ± .0121 | .0011 ± .0007 | .2082 ± .1359 |

distances and pausing of motion at shorter distances. The distance between a leader and a follower is characterized by oscillations with higher frequency but lower amplitude than those of the ants (*Figure 3A* and *Figure 4A*). These oscillations are largely within the rotation regime due to sustained motion. In this regime, tandem runners frequently alternate between a phase in which the leader is the faster of the two and their distance increases (*Figure 4C*, point 1a) and a phase in which the follower moves faster than the leader, reducing the gap (point 2a). Sporadically, leader and follower can be found very close to each other (less than 0.89 body lengths, *Figure 4B*) where they enter the pausing regime. When this happens, the leader's motion initially predicts the decrease and then the increase in speed of the follower (points 1b and 2b). The pausing regime is then quickly abandoned, and rotation information regains dominance. This behavior is consistent with relatively close proximity facilitating momentary large course corrections (*Figure 4A*, right inset). Leader-initiated pauses in termites might serve some unknown function, for example motor planning (*Card and Dickinson, 2008*; *Hunt et al., 2016*); however, unlike the case of ants, we uncover no evidence that the termite pauses facilitate follower control over any aspects of the trajectory.

## Discussion

Although both ants and termites have similar mechanisms for mutual signaling at short time scales, route learning by ants requires a communication protocol at intermediate time scales different from that needed solely to maintain spatial cohesion. In principle, both ant and termite followers can transfer information to their leaders by signaling their presence through physical contact. However, in contrast to ants, termite followers in our experiments transfer information only when establishing contact at the beginning of a run and sporadically after accidental breaks. Once contact is established and the run is proceeding steadily, termite followers cease to transfer information to their leaders who instead control both the direction and the speed of the run. Although manipulation experiments acting at short time scales can show bidirectional flow in all three species, we found evidence that communication at intermediate time scales is consistently bidirectional only in ants (from leader to follower for rotations and from follower to leader for pauses) whereas, with the exception of accidental breaks, it is consistently unidirectional in termites (from leader to follower for both rotations and pauses).

The communication protocol followed by termites can be likened to a person leading another by the hand. The protocol of ants reveals instead a more complex coordination of social behavior as leader and follower systematically alternate between close contact and separation. We suggest that the ants' intermittent motion and bidirectional feedback is akin to the pausing for acknowledgment observed between machines on a computer network. In this case, communication theory can aid in understanding the frequency of acknowledgments in terms of the receiver's informational capacity and the complexity of the information being received (*Cover and Thomas, 2005*). The selective

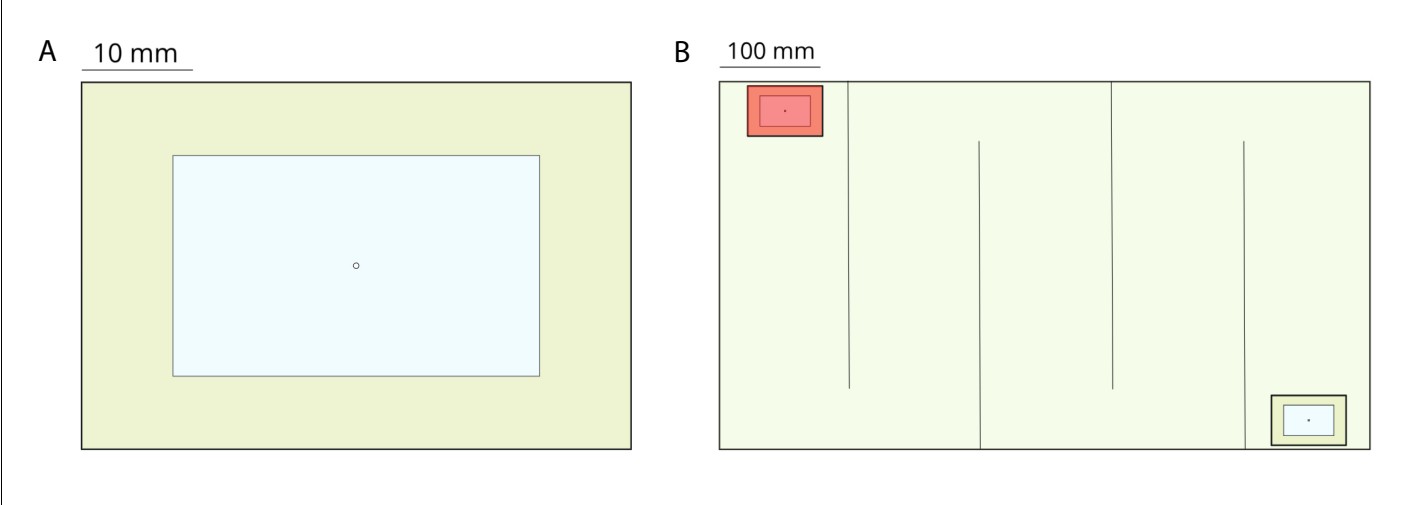

**Figure 5.** Experimental setup for experiments performed with the ant *T. rugatulus*. (**A**) Nest architecture with the entrance in the center of the roof. (**B**) Experimental arena partitioned into a corridor with the old nest (bottom right) and the new nest (top left) positioned at the extremities.

exposure of a follower to navigational information from the environment is akin to the sending of a complex message over a simple channel which, in machine-to-machine communication, requires flow control mechanisms to prevent overwhelming the receiver.

Tandem running by the related and behaviorally similar ant *T. albipennis* has also been likened to teaching—a social behavior often used to distinguish humans from other animals—because the leader modifies her behavior in the presence of a naïve follower at some cost and as a result of bidirectional feedback (*Franks and Richardson, 2006*; *Richardson et al., 2007*). Our results reveal how the regulation of information flow might be an underappreciated requirement of teaching. This assumption could be investigated by applying the methodology we put forward here to other examples of teaching known in the animal kingdom (*Hoppitt et al., 2008*). Moreover, although social insects use cue-based mechanisms for flow and congestion control of physical quantities, such as food or nesting material (*Seeley, 1989*; *Prabhakar et al., 2012*), our study is the first to reveal a protocol for the application of stereotyped signals to control the flow of other information (*i.e.*, a nonphysical quantity) in a non-human organism. Furthermore, tandem running has evolved multiple times in the ants but not all instances necessarily require acknowledgment signals (*Kaur et al., 2017*), and so comparing across taxa may reveal the ecological context that led to the evolution of signals that regulate other signals.

## Conclusions

Temporal correlations manifesting over intermediate time scales represent an important but often neglected aspect in behavioral ecology (*Mitchell et al., 2019*). Complex spatiotemporal interactions among individuals (*i.e.*, those interactions evolving over intermediate time scales) are difficult to study by direct manipulation in highly controlled laboratory settings. Instead, quantitative and noninterventional methods applied over longer observational periods can be used to capture the dynamical aspect of social interactions, but these methods are generally underdeveloped and sporadically used in behavioral ecology. As we have shown in this study, information theory offers tools such as transfer entropy that can disentangle the temporal structure of the interaction between individuals.

Whereas the construct of transfer entropy has seen extensive applications in the neurosciences, particularly to study effective connectivity in the brain (*Vicente et al., 2011*), its application in the field of animal behavior is less frequent and has focused primarily on revealing leader–follower relationships in fish (*Butail et al., 2016*; *Kim et al., 2018*; *Ward et al., 2018*) and bats (*Orange and Abaid, 2015*), with more recent applications in the study of decision-making in humans (*Grabow et al., 2016*; *Porfiri et al., 2019*) and slime molds (*Ray et al., 2019*). These previous applications set out to answer the generic question of whether one subject is influencing another on the

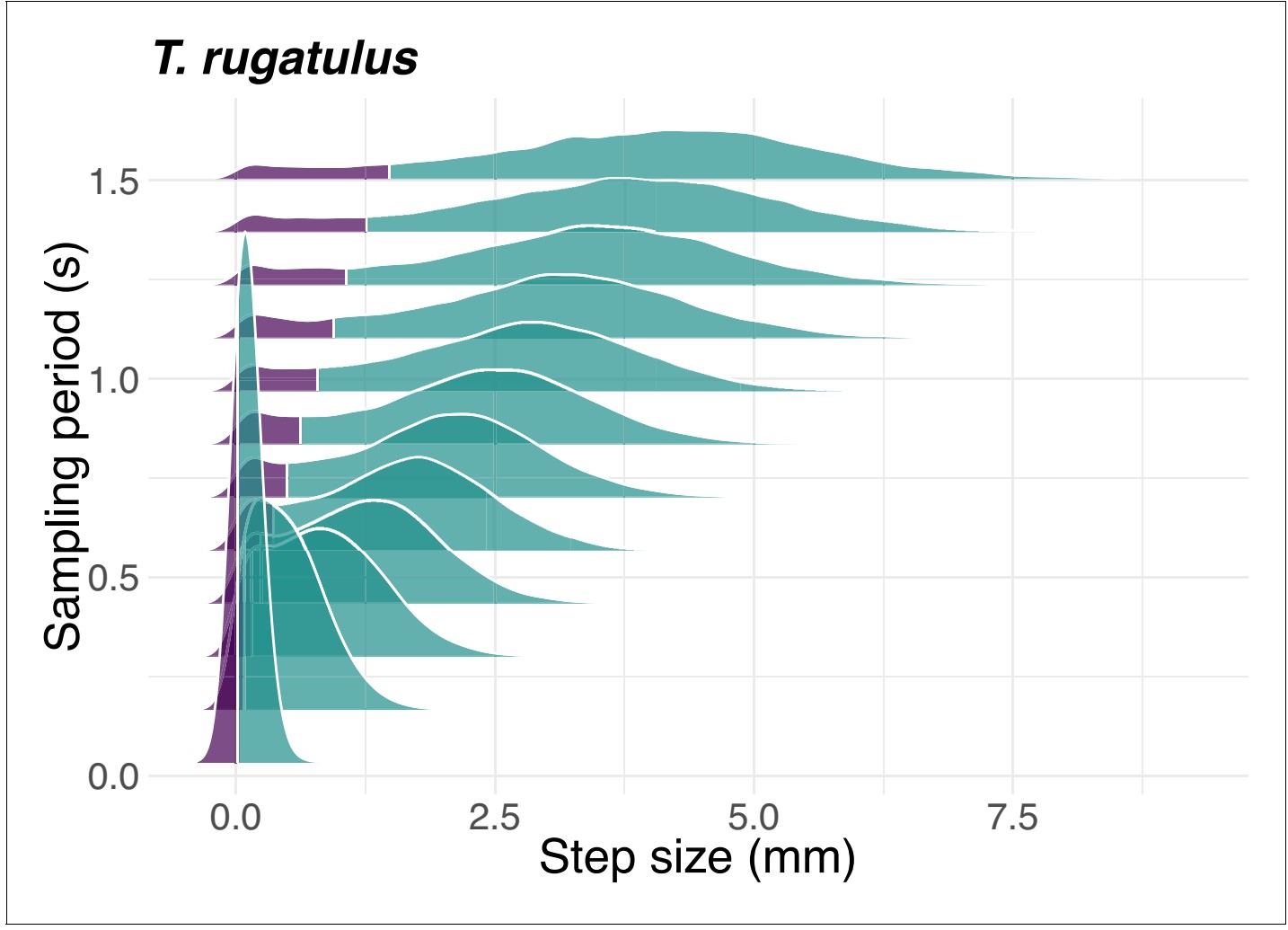

**Figure 6.** Step-size distribution as a function of sampling period. Probability density function of the step size for the ant *T. rugatulus* as a function of the sampling period. Purple represents the 10% probability mass used to define the pause state; green represents the remaining 90% of the probability mass defining the motion state. See also *Figure 6—figure supplement 1* for the termite *C. formosanus* and *Figure 6—figure supplement 2* for *R. speratus*.

The online version of this article includes the following figure supplement(s) for figure 6:

**Figure supplement 1.** Step-size distribution as a function of sampling period.

**Figure supplement 2.** Step-size distribution as a function of sampling period.

basis of a single symbolic representation of raw data (*e.g.*, a single encoding that carries information only about the direction of motion). Such uses of transfer entropy and other information-theoretic measures (*McCowan et al., 1999*) cannot disentangle the complex structure of information flow between subjects when simultaneous aspects of their interaction carry different forms of information transmitted in different directions and at different time scales. By quantifying multiple, concurrent symbolic patterns (*i.e.*, variation over time in both direction and speed) and subsequently relating the structure of information flows latent in these data to each other, we have shown how to uncover more complex communication protocols as opposed to simply identifying distinct individuals within a social interaction.

The methodology we put forward, which applies advanced information-theoretic measures to different symbolic representations of the same dataset, has allowed us to show differences in the communication protocol used by tandem-running ants and termites, and to explain the disparity in their function. Our approach is sufficiently generic to enable the discovery of cryptic signaling behaviors in other taxa and to provide deeper insights into behaviors whose function is poorly or partially

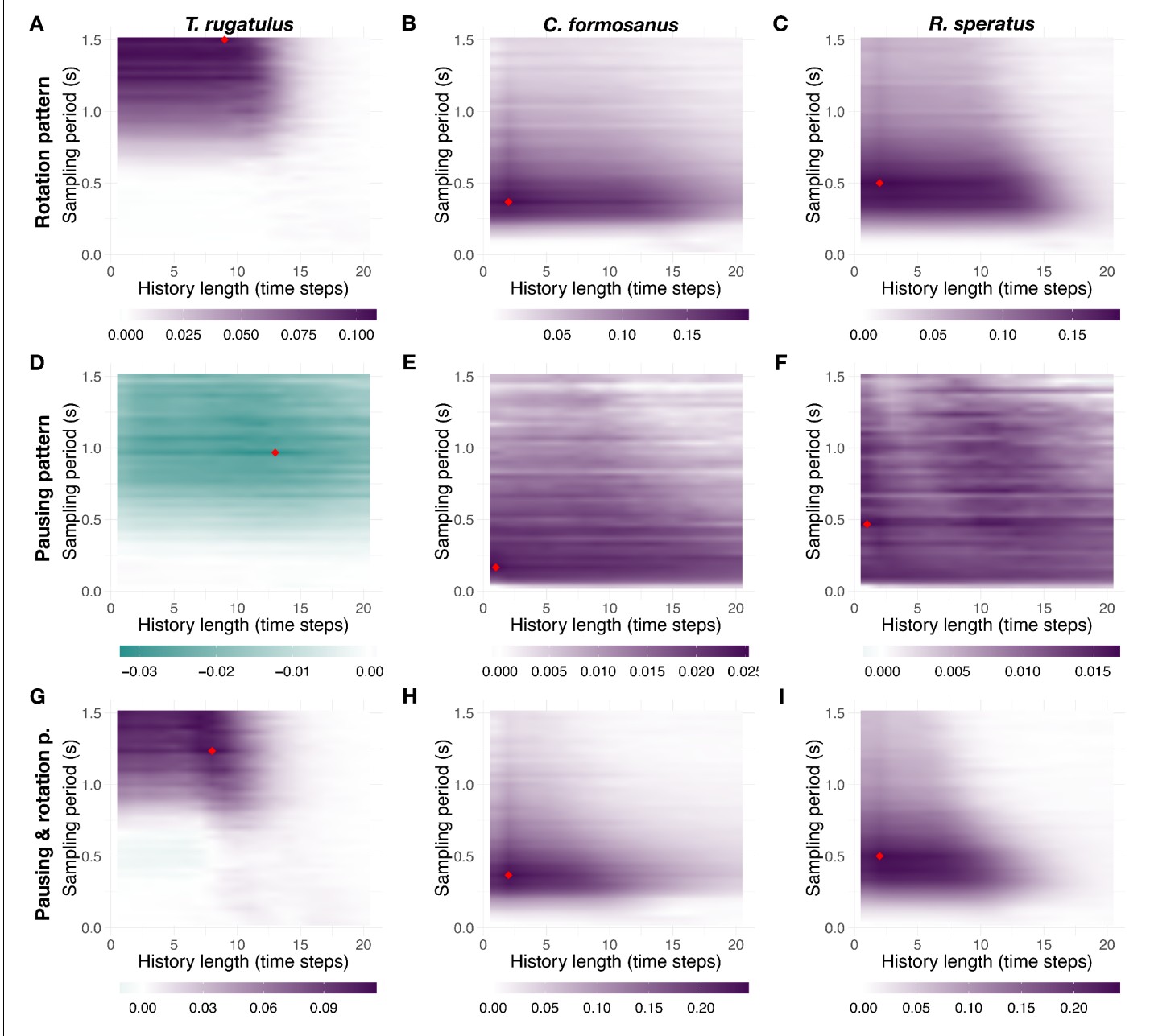

**Figure 7.** Landscape of net information transfer. Net transfer entropy (bits) as a function of the sampling period and of the history length. (A), (B), and (C) show the results for the rotation pattern, (D), (E), and (F) show the results for the pausing pattern, and (G), (H), and (I) show the results for the compound pausing and rotation pattern. The first, second, and third columns show the results, respectively, for *T. rugatulus*, *C. formosanus*, and *R. speratus*. Colors indicate the intensity and predominant direction of information transfer (purple, from leader to follower; green, from follower to leader); the red diamond symbol indicates the configuration with maximum magnitude. See also *Figure 7—figure supplements 1–3*. Source data of net transfer entropy are available in *Figure 7—source data 1*.

The online version of this article includes the following source data and figure supplement(s) for figure 7:

**Source data 1.** Transfer entropy as a function of the sampling period and of the history length.
**Figure supplement 1.** Impact of amount of data on the landscape information transfer.
**Figure supplement 2.** Perturbation analysis of pause-motion threshold.
**Figure supplement 3.** Predictive information over distance between runners.

understood (*e.g.*, turn-taking [*Flack, 2013*; *Pika et al., 2018*] and complex coordinated dances in birds [*Ota et al., 2015*]). Furthermore, we have shown how the generality of this approach can extend traditional information-theoretic analysis from a mechanistic focus on one species toward a comparison across a wide taxonomic range. Such a common language of information processing can enable the posing of new questions, hypotheses, and predictions for the evolution of information processing itself.

# Materials and methods

Data and source code are available in *Valentini et al., 2020*.

## Ant experiments

We used 6 colonies of *T. rugatulus* (between 30–60 individuals each) collected in the Pinal Mountains near Globe, Arizona, during September 2017. Each colony was kept in a plastic box (110 mm by 110 mm) with a nest, a water tube, and an agar-based diet (*Bhatkar and Whitcomb, 1970*). Nests (50 mm by 75 mm) were composed of a balsa-wood slat with a central rectangular cavity (30 mm by 50 mm) and sandwiched between two glass slides (*Figure 5A*). The top slide had a 2 mm hole over the center of the nest cavity to allow ants to enter and leave the nest. We conducted emigration experiments to induce ants to perform tandem runs. To obtain sufficiently long tandem runs, we used a large experimental arena (370 mm by 655 mm) delimited by walls (37 mm tall) and subdivided by five barriers (10 mm by 310 mm) placed to form a contiguous corridor with alternating left and right turns (*Figure 5B*). The design and dimensions of this arena were informed by a preliminary analysis of termite experiments (see Computation of statistics). Both walls and barriers were coated with Fluon to prevent ants from leaving the experimental arena. A new nest was placed at one extremity of the corridor and was covered with a transparent red filter to encourage the ants, which prefer dark cavities (*Franks et al., 2003*), to move in. The nest housing a colony was transferred from its plastic box and placed at the other extremity of the corridor. Colony emigration was induced by removing the top slide of the occupied nest. We performed six experiments, one for each colony, and recorded them at 30 frames per second using a video camera with 1K resolution. For each colony, we then selected between 1 and 6 pairs of ants performing tandem runs obtaining a total of 20 samples. Selected tandem runs last more than 15 min and have the same pair of ants travelling between the two nests with no or minimal interaction with other members of the colony.

## Termite experiments

Experiments with *C. formosanus* and *R. speratus* were performed as part of a study on sexually dimorphic movements of termites during mate search (*Mizumoto and Dobata, 2019*). Alates from 2 colonies of *C. formosanus* were collected in Wakayama, Japan, in June 2017; alates from 5 colonies of *R. speratus* were collected in Kyoto, Japan, in May 2017. After controlled nuptial flight experiments, termites that shed their wings were selected and used for tandem run experiments. Experiments were performed in a Petri dish (145 mm Ø) filled with moistened plaster whose surface was scraped before each trial. A female and a male termite were introduced in the experimental arena with the opportunity to tandem run for up to 1 hr. A total of 17 experiments were performed for *C. formosanus* and 20 experiments for *R. speratus* using different individuals. Tandem runs were recorded at 30 frames per second using a video camera with a resolution of 640 by 480 pixels.

## Data extraction

We extracted motion trajectories from video recordings of tandem runs by automatically tracking the position over time of leaders and followers. Motion tracking was accomplished using the UMA-Tracker software platform (*Yamanaka and Takeuchi, 2018*). Because we tracked the centroids of each runner's body, the distance between individuals was always greater than zero even when leader and follower were in contact with each other. All trajectories were sampled at 30 frames per second and shortened to a duration of 15 min. Trajectories were then converted from pixels to millimeters using a scaling factor estimated by measuring known features of the experimental arena with ImageJ (*Schneider et al., 2012*). The body size of each runner (average ± standard deviation) was measured from video recordings of the experiments using ImageJ (*T. rugatulus*: 2.34 ± 0.3 mm, *C. formosanus*: 8.89 ± 0.42 mm, *R. speratus*: 5.5 ± 0.3 mm).

**Table 3.** Results of hypothesis testing of information transfer between leaders and followers.

Columns 3 and 4 report the results of one-sided two-sample Wilcoxon rank-sum tests with continuity correction ($p$-value and $W$ statistic) testing if the experimental dataset has significantly higher transfer entropy than the surrogate one. Columns 5 and 6 report the results of one-sided paired Wilcoxon signed-rank tests with continuity correction ($p$-value and $V$ statistic) testing, respectively, if the leader is significantly more informative than the follower and vice versa. Significant $p$-values are reported in bold.

| Species | Behavioral pattern | $H_1$: $T_{L \to F} > T_{L \to F}^s$ | $H_1$: $T_{F \to L} > T_{F \to L}^s$ | $H_1$: $T_{L \to F} > T_{F \to L}$ | $H_1$: $T_{F \to L} > T_{L \to F}$ |
|---|---|---|---|---|---|
| T. rugatulus | Rotation | **$p$<.001** ($W = 380$) | $p = .51$ ($W = 200$) | **$p$<.001** ($V = 209$) | $p = 1.0$ ($V = 1$) |
| | Pausing | $p = .21$ ($W = 230$) | **$p$<.001** ($W = 388$) | $p = 1.0$ ($V = 0$) | **$p$<.001** ($V = 210$) |
| | Pausing and Rotation | **$p$<.001** ($W = 400$) | $p = .7$ ($W = 181$) | **$p$<.001** ($V = 207$) | $p = 1.0$ ($V = 3$) |
| C. formosanus | Rotation | **$p$<.001** ($W = 289$) | **$p = .002$** ($W = 226$) | **$p$<.001** ($V = 153$) | $p = 1.0$ ($V = 0$) |
| | Pausing | **$p$<.001** ($W = 272$) | **$p$<.001** ($W = 272$) | **$p = .001$** ($V = 138$) | $p = 1.0$ ($V = 15$) |
| | Pausing and Rotation | **$p$<.001** ($W = 289$) | **$p = .001$** ($W = 229$) | **$p$<.001** ($V = 153$) | $p = 1.0$ ($V = 0$) |
| R. speratus | Rotation | **$p$<.001** ($W = 380$) | $p = .3$ ($W = 220$) | **$p$<.001** ($V = 201$) | $p = 1.0$ ($V = 9$) |
| | Pausing | **$p$<.001** ($W = 380$) | **$p$<.001** ($W = 340$) | **$p$<.001** ($V = 195$) | $p = 1.0$ ($V = 15$) |
| | Pausing and Rotation | **$p$<.001** ($W = 380$) | **$p = .018$** ($W = 278$) | **$p$<.001** ($V = 193$) | $p = 1.0$ ($V = 17$) |

## Encoding behavioral patterns into symbolic time series

We considered three possible behavioral patterns for each runner: *pausing pattern*, *rotation pattern*, and their combination *pausing and rotation pattern*. We did so by coarse-graining the space-continuous trajectories of each leader and each follower using three different symbolic representations. Each spatial trajectory consists of a sequence $(q_1, q_2, \ldots)$ of 2-dimensional points, $q_i = (q_i^x, q_i^y)$, representing spatial coordinates over time which are then encoded into a symbolic time series $X = (x_1, x_2, \ldots)$. To capture the time interval where the sender best predicts the behavior of the receiver, we subsampled spatial trajectories in time before encoding the behavioral patterns of each runner. We considered different sampling periods, starting from a short period of one sample every 33.3667 ms (29.97 Hz) to a long period of one sample every 1.5015 s (0.666 Hz) with an interval between each period of 33.3667 ms (*i.e.*, sampling period $\in \{0.0334s, 0.667s, \ldots, 1.5015s\}$). Depending on the sampling period, the resulting time series have a number of time steps between 599 and 26973.

The pausing pattern is encoded using two states: the *motion state* (M) and the *pause* state (P). The motivation for this coding scheme is to capture when a tandem runner pauses while waiting for the other to re-join the tandem run or to react to physical contact. Pauses, small adjustments of the position of the runner, or changes due to noise in the sampled trajectories may each accidentally be considered as genuine acts of motion. To prevent these spurious classifications, we used a threshold based on travelled distance to distinguish segments of the trajectory into those identifying motion and those identifying pauses. We first computed the probability distribution of step sizes, that is, the distance travelled by a runner between two consecutive sampled positions $q_i$ and $q_{i+1}$ for each species and sampling period (*Figure 6* and supplements). These distributions show two distinct modes: short steps (*i.e.*, low speed) characteristic of pauses and long steps (*i.e.*, high speed) characteristic of sustained motion. After inspection, we chose the 10$^{th}$ percentile of each probability distribution as the distance threshold used to distinguish between motion and pauses. We therefore encoded as pause states all time steps in a given spatial trajectory with a corresponding travelled distance in the 10$^{th}$ size percentile and the remaining 90% of time steps as motion states. This threshold was varied in the interval $\{5\%, 6\%, \ldots, 15\%\}$ during a perturbation analysis of predictive information (see Computation of statistics).

The rotation pattern is also encoded using two states: *clockwise* (CW) and *counterclockwise* (CCW). The direction of rotation at time $i$ is obtained by looking at three consecutive positions, $q_{i-1}$, $q_i$, $q_{i+1}$, in the spatial trajectory of each runner. The rotation is clockwise when the cross product $\overrightarrow{q_{i-1}q_i} \times \overrightarrow{q_iq_{i+1}}$ is negative, counterclockwise when it is positive, and collinear when it is exactly zero. As we aim to model only clockwise and counter-clockwise rotations, we do not consider any

tolerance threshold to explicitly capture collinear motion. Instead, in the rare occurrences of collinear motion, the direction of rotation at the previous time step, $i - 1$, is copied over in the time series.

As a control for our choices of possible behavioral outcomes, we also considered a compound pausing and rotation pattern that simultaneously encodes both components of tandem running. A possible approach to do so is to use the time series of each behavioral pattern separately but then rely on multivariate measures of predictive information. However, in our scenario, pauses have a mutually exclusive relation with rotations that cannot be preserved using multivariate measures. The pausing and rotation pattern is defined therefore using a ternary coding scheme that encodes motion bouts in the states *pause* (P), *clockwise* (CW) and *counterclockwise* (CCW). As for the pausing pattern, the shortest 10% of steps in the spatial trajectories are encoded as pausing (see Computation of statistics for a perturbation analysis of this parameter). The remaining 90% of steps are encoded using states clockwise and counterclockwise following the same methodology used for the rotation pattern.

## Measuring predictive information

Our analysis of communication in tandem running is grounded in the theory of information (***Cover and Thomas, 2005***) and its constructs of entropy, conditional entropy, and transfer entropy. We aim to quantify how knowledge of the current behavior of the sender allows us to predict the future behavior of the receiver, that is, to measure causal interactions in a Wiener-Granger sense (***Bossomaier et al., 2016***). We consider the behavioral patterns of leaders and followers as the series of realizations $(l_i, i \geq 1)$ and $(f_i, i \geq 1)$ of two random variables, $L$ and $F$, corresponding to the leader and follower, respectively. For simplicity, the following presentation focuses on predicting the future of the follower, $F^{i+1} = (f_{i+1}, i \geq 1)$, from the present of the leader, $L$, but in our analysis we also consider how much of the future of the leader, $L^{i+1}$, is predicted by the present of the follower, $F$.

The overall uncertainty about the future $F^{i+1}$ of the follower is quantified by the (marginal) entropy (***Shannon, 1948***) $H(F^{i+1}) = -\sum_{f_{i+1}} p(f_{i+1}) log_2 p(f_{i+1})$. Entropy measures the average amount of information necessary to uniquely identify an outcome of $F^{i+1}$. Knowing the history of the follower may reduce the uncertainty in the distribution of possible outcomes for the future of the follower, and the reduction in uncertainty can be quantified by the difference between the marginal entropy and the entropy after the historical information is considered. Let $f_i^{(k)} = \{f_{i-k+1}, \ldots, f_{i-1}, f_i\}$ represent the finite history with length $k$ of $F$ up to the current time $i$ and $F^{(k)}$ a new random variable defined over a series $\left(f_i^{(k)}, i \geq 1\right)$ of $k$-histories. The amount of uncertainty about $F^{i+1}$ that is left after accounting for its past behavior $F^{(k)}$ is given by the conditional entropy:

$$H\left(F^{i+1}|F^{(k)}\right) = -\sum_{f_i^{(k)}, f_{i+1}} p\left(f_i^{(k)}, f_{i+1}\right) log_2 \frac{p\left(f_i^{(k)}, f_{i+1}\right)}{p\left(f_i^{(k)}\right)},$$

for history length $1 \leq k < \infty$. $H\left(F^{i+1}|F^{(k)}\right)$ represents the average amount of information necessary to uniquely identify the future behavior of the follower given what we know about its past behavior.

A second step to obtain additional information about the future of the follower is to consider the time-delayed effects of its interaction with the leader. Transfer entropy was introduced for this purpose (***Schreiber, 2000***). It measures the amount of information about the future behavior of the receiver given by knowledge of the current behavior of the sender that is not contained in the receiver's past. Due to its time directionality (*i.e.*, from the present of the sender to the future of the receiver), it is considered a measure of information transfer or predictive information (***Lizier and Prokopenko, 2010***). Transfer entropy is defined as:

$$T_{L \to F} = \sum_{f_{i+1}, f_i^{(k)}, l_i} p\left(f_{i+1}, f_i^{(k)}, l_i\right) log_2 \frac{p\left(f_{i+1} | f_i^{(k)}, l_i\right)}{p\left(f_{i+1} | f_i^{(k)}\right)}$$

and measures the reduction of uncertainty of $F^{i+1}$ given from knowledge of $L$ which is not already given by $F^{(k)}$. The logarithm in the above equation is known as local transfer entropy (***Lizier et al.,***

*2008*) and tells us whether, at time $i$, the interaction $l_i \, | \, f_i^{(k)} \rightarrow f_{i+1} \, | \, f_i^{(k)}$ between the two processes is informative (>0) or misinformative (<0). In our analysis, we look at local transfer entropy averaged over the distance between leader and follower to understand the spatiotemporal dynamics of communication during tandem running.

Due to the asymmetry of transfer entropy, $T_{L \rightarrow F} \neq T_{F \rightarrow L}$, we can obtain the predominant direction and the magnitude of predictive information by studying the difference:

$$T_{L \rightarrow F} - T_{F \rightarrow L}.$$

This quantity is positive when information flows predominantly from leader to follower ($T_{L \rightarrow F} > T_{F \rightarrow L}$) and negative when it flows from follower to leader ($T_{L \rightarrow F} < T_{F \rightarrow L}$). Its value is known as net transfer entropy (*Porfiri, 2018*). Finally, as transfer entropy can be rewritten as $T_{L \rightarrow F} = H(F^{i+1} | F^{(k)}) - H(F^{i+1} | F^{(k)}, L)$, we can normalize this quantity in the interval $[0; 1]$ simply by dividing it by the conditional entropy as in:

$$\frac{T_{L \rightarrow F}}{H(F^{i+1} | F^{(k)})} = \frac{H(F^{i+1} | F^{(k)}) - H(F^{i+1} | F^{(k)}, L)}{H(F^{i+1} | F^{(k)})}.$$

Normalized transfer entropy (*Porfiri, 2018*) is a dimensionless quantity that captures the proportion of the future behavior $F^{i+1}$ of the follower that is explained by the interaction with the leader at time $i$. When $F^{i+1}$ is completely predicted by $L$, the conditional entropy $H(F^{i+1} | F^{(k)}, L)$ is zero and normalized transfer entropy is maximal and equal to 1; instead, when $F^{i+1}$ is independent of $L$, $H(F^{i+1} | F^{(k)}) = H(F^{i+1} | F^{(k)}, L)$ and normalized transfer entropy is minimal and equal to 0.

## Computation of statistics

We computed information-theoretic measures for both leaders and followers. In our computations, we assume that the pausing and rotation patterns of ants and termites are peculiar features of the species rather than of specific pairs of tandem runners. As such, rather than treating each trial separately and then aggregating the results, we estimated the necessary probabilities from all experimental trials together and obtained a single estimate of transfer entropy for each considered species and parameter configuration. Our measures of predictive information are therefore averaged over all trials of the same species. Probability distributions are estimated from the frequencies of blocks of consecutive symbols within the time series. For example, the probability for a follower to have a history of rotations $f_i^{(3)} = \{f_{i-2} = CW, f_{i-1} = CW, f_i = CCW\}$ is estimated by counting the number of times the symbols $\{CW, \ CW, \ CCW\}$ occur consecutively at any point in the time series of any follower; this count is then normalized by the number of samples to obtain a measure of probability. All information-theoretic measures were computed in R 3.4.3 using the rinform-1.0.1 package (*Moore et al., 2018*).

To ensure that the measured interactions are valid and not the result of artefacts that may arise due to finite sample sets, we compared transfer entropy measured from the experimental data with measurements from surrogate datasets artificially created by pairing independent time series (*Porfiri, 2018*). To create a surrogate dataset, we randomly paired the behavioral patterns of leaders and followers belonging to different tandem runs, obtaining a dataset with the same size as the original. We then computed transfer entropy for this surrogate data. Although leaders and followers from different runs are still influenced by the same environmental cues, this randomization process breaks possible causal interactions within the surrogate pair. For each species and parameter configuration, we repeated this randomization process 50 times obtaining 50 surrogate datasets that were used to estimate mean and standard error of transfer entropy. Finally, measurements of transfer entropy for the experimental data were discounted by a correction factor given by the estimated means.

The sampling period of continuous spatial trajectories and the history length of transfer entropy define the parameter space of our study. The optimal choice of these parameters likely varies for different species and between leaders and followers within a species as a result of behavioral, morphological, and cognitive differences manifesting at different time scales. To choose a suitable parameter configuration and control for its robustness, we computed net transfer entropy for 900 different parameter configurations for each species (history length $k \in \{1, \ldots, 20\}$ and sampling

period $\{0.0334s, \dots, 1.5015s\}$). From the resulting landscapes of information transfer, which show robustness to variation of parameters, we then selected the parameter configurations that maximize the net transfer of information (see *Figure 7* and *Table 1*). A similar analysis was preliminarily performed for increasing duration of tandem runs on the basis of the termite data with the aim to inform the design of the experimental arena used for ants (see *Figure 7—figure supplement 1*). These parameters, whose values converge for increasing length of the time series, are relatively similar across behavioral patterns for both species of termites. Ants instead are characterized by more diverse time scales likely because leaders and followers cause different aspects of tandem running and, possibly, because they do so by following cognitive processes with different time constraints (see *Table 2* for summary statistics).

For the chosen parameter configurations, we tested both the significance of our estimate of transfer entropy with respect to surrogate data and that of leader–follower relations observed in *Figure 2A*. One-sided two-sample Wilcoxon rank-sum tests with continuity correction show values of transfer entropy for the experimental data significantly greater than those for surrogate data (*Table 3*, columns 3 and 4) in all but four of these tests. One-sided paired Wilcoxon signed-rank tests with continuity correction were used instead to test differences in causal interactions between leaders and followers and to confirm the effects shown in *Figure 2A*. All leader–follower interactions were correctly identified by this analysis and none of the significant tests is among the four cases mentioned above (*Table 3*, columns 5 and 6). Next, we performed a perturbation analysis of the probability threshold used to separate pauses from motion in the pausing pattern and in the pausing and rotation pattern ($\{5\%, 6\%, \dots, 15\%\}$). Although the magnitude is subject to some variation, the direction of information transfer that represents our primary observable remains unaltered (see *Figure 7—figure supplement 2*). Finally, we also controlled for our choices of symbolic representation on possible outcomes in the behavioral patterns by considering a compound pausing and rotation pattern. *Figure 7—figure supplement 3* shows the results of this analysis which closely resemble those shown in *Figure 3* for *T. rugatulus*, *Figure 4* for *C. formosanus*, and *Figure 4—figure supplement 1* for *R. speratus*.

## Acknowledgements

This work was supported by NSF grant No. PHY-1505048. NM was supported by a JSPS Overseas Research Fellowship. The authors would like to thank Dr. Anna Dornhaus for constructive criticism of the manuscript.

## Additional information

### Funding

| Funder | Grant reference number | Author |
| --- | --- | --- |
| National Science Foundation | PHY-1505048 | Stephen Pratt<br>Theodore P. Pavlic<br>Sara Walker |
| Japan Society for the Promotion of Science | Overseas Research Fellowship | Nobuaki Mizumoto |

The funders had no role in study design, data collection and interpretation, or the decision to submit the work for publication.

### Author contributions

Gabriele Valentini, Conceptualization, Data curation, Formal analysis, Investigation, Visualization, Methodology, Writing - original draft, Writing - review and editing; Nobuaki Mizumoto, Investigation, Writing - review and editing; Stephen C Pratt, Conceptualization, Supervision, Funding acquisition, Writing - review and editing; Theodore P Pavlic, Conceptualization, Supervision, Funding acquisition, Writing - original draft, Writing - review and editing; Sara I Walker, Conceptualization, Supervision, Funding acquisition, Project administration, Writing - review and editing

## Author ORCIDs

Gabriele Valentini [ID] https://orcid.org/0000-0002-8961-3211
Nobuaki Mizumoto [ID] https://orcid.org/0000-0002-6731-8684
Stephen C Pratt [ID] https://orcid.org/0000-0002-1086-4019
Theodore P Pavlic [ID] https://orcid.org/0000-0002-7073-6932
Sara I Walker [ID] https://orcid.org/0000-0001-5779-2772

## Decision letter and Author response

Decision letter https://doi.org/10.7554/eLife.55395.sa1
Author response https://doi.org/10.7554/eLife.55395.sa2

## Additional files

### Supplementary files

• Transparent reporting form

### Data availability

Data and code is available in Figshare. 2020. https://doi.org/10.6084/m9.figshare.9786260.

The following dataset was generated:

| Author(s) | Year | Dataset title | Dataset URL | Database and Identifier |
|---|---|---|---|---|
| Valentini G, Mizumoto N, Pratt S, Pavlic TP, Walker SI | 2020 | Data and Code from: Revealing the Structure of Information Flows Discriminates Similar Animal Social Behaviors | https://doi.org/10.6084/m9.figshare.9786260 | figshare, 10.6084/m9.figshare.9786260 |

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
