## [Decision Letter]

**Acceptance summary:**

Your analyses of social interaction with information theory are very interesting and pave a new way for the study of the organisation of social organisms. We are therefore pleased to publish your paper.

**Decision letter after peer review:**

Thank you for submitting your article "Revealing the structure of information flows discriminates similar animal social behaviors" for consideration by *eLife*. Your article has been reviewed by three peer reviewers, and the evaluation has been overseen by a Reviewing Editor and Christian Rutz as the Senior Editor. The following individual involved in the review of your submission has agreed to reveal their identity: Jean-Louis Deneubourg (Reviewer #1).

On the basis of the three reviews, which are appended below, the Reviewing Editor has drafted this decision letter to help you prepare a revised submission. In recognition of the fact that revisions may take longer than the two months we typically allow, until the research enterprise restarts in full, we will give authors as much time as they require to submit revised manuscripts.

We agree with the three reviewers that this well-written paper presents interesting work that should be of interest to a general audience. The three reviewers made valuable comments, which all need to be addressed carefully. We agree in particular with their concern that it is not easy to make robust comparisons across species, given differences in their biology and the experimental procedures used, so this point requires particular attention.

Reviewer #1:

The Valentini et al.'s work is highly original and opens new perspectives for better understanding how animal communicate. This paper presents highly original methods (based on the information theory) and results (interaction between social insects) as to justify its publication in *eLife*. In short, I strongly recommend its publication.

Valentini et al. show that non-invasive information-theoretic tools reveal the communication protocol by measuring simultaneous flows of different information between social individuals. The authors demonstrate the power of their approach/method by showing that the tandem recruitment of ants and termites are governed by different communication protocols. They also stress – and I share their statements – the fact that their method is non-invasive (based only on observational data from many repeated interactions) and does not rely on a priori assumptions (model-free). The subject is not only an issue concerning biology of social insects but it is of interest to a wide readership including specialists in social behaviour, collective phenomena, complex systems, etc.

More specifically, one of the interesting results – and it is surprising – is the difference concerning the transfer of information (in tandem recruitment) from follower to leader between ants and termites.

The Abstract summarizes well the content of the paper. The Introduction does a good job summarizing the biological challenges (function of tandem running behaviour) and the theoretical background (information theory). The manuscript is very well-written and very pleasant to read. The goals of the work are clearly set out, the methods (and theoretical tools) are very well explained and the results are impressive and convincing.

Reviewer #2:

This paper provides an information-theoretic explanation for the different purposes of tandem running in ants and termites. In agreement with the literature, while followers in ants participate in the tandem runs to learn a route, and followers in termites simply help explore the region the authors reveal different directionalities of information transfer between these species. These different purposes manifest in the form of how leader-follower relationships emerge within seemingly similar trajectory data. Symbolic time-series representation is used to encode different behaviors within the tandem run. The paper is well written and the results are impactful providing a mathematical interpretation to complex behaviors. The authors have taken care to ensure that the results are robust through a detailed sensitivity analysis.

I have few questions/comments regarding the methodology:

1) I was not able to understand how exactly the behaviors were encoded? Is there a manual step or is it all automatic. The para starting from 267 gets somewhat involved and does not explain how exactly the authors went from trajectory data to symbols. What exactly do the authors mean by 10^th^ percentile of a sequence. The next paragraph on rotation was clearer but there too I was wondering if there were any thresholds to distinguish collinear motion from rotational motion.

2) Related to above, how did the authors determine the length of a sequence to be encoded? Was it a moving window?

3) What was the size of the dataset? How many instances of behaviors were compared for each species?

4) Is the leader ant always the one moving in front. This should be explicitly noted somewhere because it may not be true for all species. Perhaps the authors can consider renaming front-back ants/termites to not confuse with leader-follower which term is use to imply a causal relationship in itself.

5) What were the reasons for selecting different sampling periods and history lengths for the same species. For the ant for example the values are very far apart. The authors can either (a) pick similar values where they can, or (b) discuss why they think information transfer is maximized at different scales for the same species for different behaviors.

6) I couldn't find any statistics on the rotation bars in Figure 1D.

Reviewer #3:

The paper demonstrates an interesting model-free methodology for extracting and interpreting qualitatively different behaviors in pairs of insects moving together in a nominally similar way (i.e. tandem running). The experiments and analytical tools are generally well explained. However, I have a fundamental concern about the conclusions that are drawn as they relate to the species tested, but not acknowledging the dramatic differences in the experimental procedures used for ants versus termites.

1) My major concern is that the experiments used to gather the data from the ants and the two species of termites seem very different. The ants are recorded when the whole colony is perturbed and seeks to move to a new nest. The termites are sexed pairs exploring their environment in a featureless petri dish. I am not an expert in social insects, but I would imagine either one of these species would behave differently in the two different experiments. I think a strong argument must be made if that's not the case. If that is the case, I think the work here is still very valuable. However, the conclusion which can be drawn from the results may be not so much about differences between species, but instead about how the information-theoretic methodology is able to extract qualitatively different strategies that the animals may be using.

2) Another concern is the lack of detail provided about the correction factor used as a control condition. The authors scramble the data from matched pairs of insects and rerun the analysis to measure a "baseline" information flow which simply is the result of these animals moving in the same space and responding to the same global cues. This is a really important part of the study in my opinion, since it is known that transfer entropy cannot distinguish between "x causes y" and "z causes x and y". The correction factor which is computed should be reported, and statistical tests should be performed to say that the measured effect is significantly more than this control. Perhaps this was done by the authors, but I was not able to tell from the text.

---

## [Author Response]

We agree with the three reviewers that this well-written paper presents interesting work that should be of interest to a general audience. The three reviewers made valuable comments, which all need to be addressed carefully. We agree in particular with their concern that it is not easy to make robust comparisons across species, given differences in their biology and the experimental procedures used, so this point requires particular attention.

We thank the editors and all the reviewers for their constructive feedback that helped us to improve our study. We revised the manuscript and hope you will find the revised version suitable for publication in *eLife*. In our response to the reviewers (see below), we provide detailed descriptions of how we answered each of the reviewers’ comments reporting all main edits to the manuscript. Among the uploaded files, you can find a version of the manuscript with tracked changes.

In the current version, we put particular emphasis on clarifying differences in the function, and therefore in the experimental protocol, between ants and termites. We are also more forward-looking regarding the fact that our results are peculiar to these differences rather than generic across species of ants and termites (see the answer to comment #2 of reviewer #3).

We also improved our reporting of statistics as suggested by the reviewers by including a table summarizing mean and standard error for various measures and all considered datasets (experimental and surrogate), a table summarizing results from statistical hypothesis tests testing the significance of all effects, and the results of a power analysis for the length of time series.

In addition to the edits necessary to address the comments of editors and reviewers, we rearranged figures and tables throughout the manuscript. The revised manuscript has 7 main figures, 3 main tables, and 6 figure supplements. Finally, we revised the material available on figshare: datasets include, in addition to raw data, all intermediate results and aggregated statistics; the source code includes functions to repeat every step of the analysis and plot all figures.

Reviewer #2:[…] I have few questions/comments regarding the methodology:1) I was not able to understand how exactly the behaviors were encoded? Is there a manual step or is it all automatic. The para starting from 267 gets somewhat involved and does not explain how exactly the authors went from trajectory data to symbols. What exactly do the authors mean by 10^th^ percentile of a sequence. The next paragraph on rotation was clearer but there too I was wondering if there were any thresholds to distinguish collinear motion from rotational motion.

Given a set of continuous spatial trajectories and a choice for the distance threshold, the encoding of all three behavioral patterns is entirely automatic (see function discretizeTrajectories() available in the *preprocessing.R* script, https://figshare.com/s/91986a474d89938e8239).

To capture pauses in the motion of runners, we first compute the probability distribution of step sizes (i.e., the distance travelled by any runner between two consecutive time steps, i and i+1, for a given species and sampling period) which is analogous to considering the instantaneous speed of runners. This provide us with a model of motion/speed that we can leverage to define pause and motion states. Specifically, we define as pause states all those time steps within a time series whose corresponding travelled distance is within the lowest 10% of the probability mass of all considered runners (10^th^ percentile). The remaining 90% of the time steps are encoded as motion states. The only manual step required was for us to initially inspect the probability distribution of step sizes (see Figure 6 and Figure 6—figure supplements 1 and 2) to determine a suitable initial value for the probability threshold before encoding time series programmatically and varying this parameter in the perturbation analysis. We clarified this aspect of the methodology in Materials and methods (Encoding behavioral patterns into symbolic time series, second paragraph):

“To prevent these spurious classifications, we used a threshold based on travelled distance to distinguish segments of the trajectory into those identifying motion and those identifying pauses. […] We therefore encoded as pause states all time steps in a given spatial trajectory with a corresponding travelled distance in the 10^th^ size percentile and the remaining 90% of time steps as motion states.”

For the case of the rotation pattern, no thresholds were used to distinguish between rotational motion (i.e., both clockwise and counterclockwise turns) and collinear motion as this encoding has a formal definition based on the cross product qi−1qi⃗×qiqi+1⃗. In particular, we have collinear motion only when qi−1qi⃗×qiqi+1⃗ is exactly zero without considering any tolerance threshold. Due to tracking accuracy and position adjustments at the highest sampling frequencies and to the likely rotations induced by a six-legged locomotion architecture at lower sampling frequencies, collinear motion is observed in our entire dataset only a handful of times (i.e., our description of the collinear case is aimed at completeness of the methodology). This aspect is advantageous for us because we want to model only rotations. We clarified this point as follows (Encoding behavioral patterns into symbolic time series, third paragraph):

“The rotation is clockwise when the cross product qi−1qi⃗×qiqi+1⃗ is negative, counterclockwise when it is positive, and collinear when it is exactly zero. […] Instead, in the rare occurrences of collinear motion, the direction of rotation at the previous time step, i−1, is copied over in the time series.”

2) Related to above, how did the authors determine the length of a sequence to be encoded? Was it a moving window?

The choice for this study parameter was the result of a compromise between the statistical power of the observed primary effects (Figure 2A and Table 3), the number of samples (i.e., time steps per series) required to accurately represent local predictive information as a function of the distance between runners (Figures 3B and 4B) which spreads samples over an additional dimension, and physical and tracking constraints of emigrations experiments with *T. rugatulus*.

As termite data were already available and their tandem runs last for about an hour, we initially used these datasets to perform the study of the landscape of information transfer as a function of sampling period and history length (i.e., equivalent to Figure 7) for increasing lengths of the time series. This procedure provided us with an educated guess for the minimum duration of tandem runs to be recorded from ant emigrations (about 10 minutes, Figure 7—figure supplement 1). Since ants cease to tandem run as soon as they reach their destination, we then designed a dedicated experimental setup (Figure 5) aimed at maximizing their total travelled distance (and time) while still allowing us to track trajectories of runners. The resulting setup led to tandem runs that were generally longer than 15 minutes. We selected 20 of such tandem runs and shortened trajectories to 15 minutes to obtain time series of equal length.

In the revised version of the manuscript we included a new figure (Figure 7—figure supplement 1) showing the results of this procedure and we added the following sentence to Materials and methods (Ant experiments):

“The design and dimensions of this arena were informed by a preliminary analysis of termite experiments (see Computation of statistics).”

We also included this description in Materials and methods (Computation of statistics, third paragraph):

“From the resulting landscapes of information transfer, which show robustness to variation of parameters, we then selected the parameter configurations that maximize the net transfer of information (see Figure 7 and Table 1). A similar analysis was preliminarily performed for increasing duration of tandem runs on the basis of the termite data with the aim to inform the design of the experimental arena used for ants (see Figure 7—figure supplement 1).”

3) What was the size of the dataset? How many instances of behaviors were compared for each species?

We acknowledge that this information was not accessible in a unique point of our manuscript as we preferred to provide these details separately in the Materials and methods where the narrative allows us to provide better explanations. For each species, we considered 20 tandem runs with the exception of *C. formosanus* for which only 17 tandem runs were available. This information can be found in Materials and methods, respectively, in subsections Ants experiments and Termites experiments:

“We performed 6 experiments, one for each colony, and recorded them at 30 frames per second using a video camera with 1K resolution. For each colony, we then selected between 1 and 6 pairs of ants performing tandem runs obtaining a total of 20 samples.”

and

“A total of 17 experiments were performed for *C. formosanus* and 20 experiments for *R. speratus* using different individuals.”

The overall duration of the times series extracted from these tandem runs is described in the section Data extraction:

“All trajectories were sampled at 30 frames per second and shortened to a duration of 15 minutes.”

As we vary the sampling period (see Figure 7), the length of time series varies for each considered parameter configuration. The maximum number of time steps in a time series is 26973 when the sampling period is the shortest (33.36 ms); the minimum number is 599 when sampling period is the longest (1.5015 s). We included the following sentence in Materials and methods (Encoding behavioral patterns into symbolic time series, first paragraph):

“Depending on the sampling period, the resulting time series have a number of time steps between 599 and 26973.”

To provide a single point in the manuscript that summarizes all parameters, we added the number of tandem runs and the exact number of time steps of time series for each selected parameter configuration in Table 1.

4) Is the leader ant always the one moving in front. This should be explicitly noted somewhere because it may not be true for all species. Perhaps the authors can consider renaming front-back ants/termites to not confuse with leader-follower which term is use to imply a causal relationship in itself.

We thank the reviewer for their suggestion. Leader and follower always refer, respectively, to the runner in the front and the one in the back of the tandem run. As this terminology has long been used within the community of researchers studying tandem running, we preferred to clarify its meaning at the beginning of the manuscript (while explicitly mentioning causal interactions at the place of leadership for the results of transfer entropy) instead of changing terminology to avoid confusion. We clarified the terminology in the Introduction of the manuscript (second paragraph) as follows:

*“*Consider the tandem-running behavior of many ants and termites, in which one individual leads a follower through their environment, the follower walking closely behind the leader throughout the run. (Figure 1A).”

5) What were the reasons for selecting different sampling periods and history lengths for the same species. For the ant for example the values are very far apart. The authors can either (a) pick similar values where they can, or (b) discuss why they think information transfer is maximized at different scales for the same species for different behaviors.

On the one hand, the reason to choose different parameter configurations is to prioritize accuracy over simplicity (less parameters) and select the parameters that, given our data, maximize transfer of information for each different behavioral pattern. Choosing parameters that maximize information transfer is the canonical approach in the case of a single symbolic representation of data as it generally provides parameterizations that better capture underlying patterns. Our approach is to apply this criterion to each different behavioral pattern. Averaging parameterizations reduces the number of parameters, simplifying the discussion to a certain extent, but at the risk of reducing the accuracy.

On the other hand, different individuals can cause different components (e.g., rotation versus motion) of a more complex behavior as observed for leaders and followers in the case of ants. Furthermore, one individual can in general exert its influence on separate components of a behavior but do this at possibly different time scales because it follows different decision-making processes and/or responds to different cues. In the case of ants, whereas both leader and follower likely respond to the same environmental cues, the cognitive requirements of learning new information (i.e., learning of visual landmarks by the follower) are likely different from those necessary to recall information already available in the brain and compare it with environmental cues (navigation by the leader).

We clarified these points in Materials and methods (Computation of statistics, third paragraph) by adding the following sentences:

“The optimal choice of these parameters likely varies for different species and between leaders and followers within a species as a result of behavioral, morphological, and cognitive differences manifesting at different time scales.”

“These parameters, whose values converge for increasing length of the time series, are relatively similar across behavioral patterns for both species of termites. Ants instead are characterized by more diverse time scales likely because leaders and followers cause different aspects of tandem running and, possibly, because they do so by following cognitive processes with different time constraints (see Table 2 for summary statistics).”

We also mentioned the possibility of different time scales in the second paragraph of the Conclusions:

“Such uses of transfer entropy and other information-theoretic measures (McCowan, Hanser and Doyle, 1999) cannot disentangle the complex structure of information flow between subjects when simultaneous aspects of their interaction carry different forms of information transmitted in different directions and at different time scales.”

6) I couldn't find any statistics on the rotation bars in Figure 1D.

In the revised version of the manuscript we included two new tables: Table 2 gives mean and standard error for transfer entropy (both experimental and control/surrogate datasets) and for normalized transfer entropy; Table 3 gives the results of hypothesis testing by reporting p-value and test statistic for tested difference between experimental and surrogate datasets and between leaders and followers.

In addition to Table 2 and Table 3, we also modified Figure 2A (formerly referred to as Figure 1D) by adding error-bars that represent the standard error characteristic of the mean of normalized transfer entropy computed over tandem runs.

Reviewer #3:The paper demonstrates an interesting model-free methodology for extracting and interpreting qualitatively different behaviors in pairs of insects moving together in a nominally similar way (i.e. tandem running). The experiments and analytical tools are generally well explained. However, I have a fundamental concern about the conclusions that are drawn as they relate to the species tested, but not acknowledging the dramatic differences in the experimental procedures used for ants versus termites.

We thank the reviewer for their constructive feedback and very much appreciate their comments. In the following, we describe how we revised the manuscript to better acknowledge differences in experimental procedures and to clarify the focus of our conclusions.

1) My major concern is that the experiments used to gather the data from the ants and the two species of termites seem very different. The ants are recorded when the whole colony is perturbed and seeks to move to a new nest. The termites are sexed pairs exploring their environment in a featureless petri dish. I am not an expert in social insects, but I would imagine either one of these species would behave differently in the two different experiments. I think a strong argument must be made if that's not the case. If that is the case, I think the work here is still very valuable. However, the conclusion which can be drawn from the results may be not so much about differences between species, but instead about how the information-theoretic methodology is able to extract qualitatively different strategies that the animals may be using.

We believe that the last sentence of the reviewer’s comment perfectly summarizes the objectives of our study. Indeed, whereas our measurements of information transfer are quantitative, our comparison between ants and termites is instead qualitative and focused on the direction of causal interactions (Figure 2) and underlying communication strategies (Figure 3 and 4).

As the reviewer suggests, the function of tandem runs is different between ants and termites. Ants use tandem runs as a recruitment mechanism to a known location whereas termites rely on this behavior when searching the environment. We are not aware of examples of ants using tandem runs as a search strategy; likewise, termites do not generally use tandem runs for recruitment, which instead relies on pheromone trails, even though there is some evidence in the workers of the most basal termites that the former behavior might have been a precursor of the latter (Sillam-Dussès, David, et al. *"*Trail-following pheromones in basal termites, with special reference to Mastotermes darwiniensis." Journal of Chemical Ecology 33.10 (2007): 1960-1977).

Differences in the function of tandem running between ants and termites would make a quantitative comparison of information transfer appropriate only between the two species of termites. However, these functional differences do not affect a qualitative comparison. In our case, they provide us instead with the primary rational to qualitatively investigate the communication strategies underlying an otherwise superficially similar behavior.

In the revised manuscript, we clarified the different contexts of tandem running in the second paragraph of the Introduction:

“At long time scales, however, there exist clear functional differences in these seemingly similar behaviors. […] In a mated pair, the male follows the female leader only to maintain spatial cohesion when searching for a new home; once a suitable location is found, the termites remain there to start a new colony, and neither partner ever retraces the route of their tandem run.”

We also clarified at the beginning of the Discussion that our results are peculiar of the different contexts and functions of tandem running across the considered species of ants and termites.

“Although both ants and termites have similar mechanisms for mutual signaling at short time scales, route learning by ants requires a communication protocol at intermediate time scales different from that needed solely to maintain spatial cohesion. […] However, in contrast to ants, termite followers in our experiments transfer information only when establishing contact at the beginning of a run and sporadically after accidental breaks.”

2) Another concern is the lack of detail provided about the correction factor used as a control condition. The authors scramble the data from matched pairs of insects and rerun the analysis to measure a "baseline" information flow which simply is the result of these animals moving in the same space and responding to the same global cues. This is a really important part of the study in my opinion, since it is known that transfer entropy cannot distinguish between "x causes y" and "z causes x and y". The correction factor which is computed should be reported, and statistical tests should be performed to say that the measured effect is significantly more than this control. Perhaps this was done by the authors, but I was not able to tell from the text.

We thank the reviewer for their suggestion. Although leaders and followers from different runs are still influenced by the same environmental cues, the randomization process used to compute the correction factor breaks possible causal interactions within the surrogate pair. We revised the corresponding description in the manuscript increasing the level of detail and including the results of statistical hypothesis tests (Table 3). We now refer explicitly to surrogate datasets and provide statistics for mean values and their standard errors (Table 2). The revised explanation of surrogate dataset and correction factor in Materials and methods (Computation of statistics, second paragraph) is:

“To ensure that the measured interactions are valid and not the result of artefacts that may arise due to finite sample sets, we compared transfer entropy measured from the experimental data with measurements from surrogate datasets artificially created by pairing independent time series (Porfiri, 2018). […] Finally, measurements of transfer entropy for the experimental data were discounted by a correction factor given by the estimated means.”

In the last paragraph of Computation of statistics we now report the results of statistical hypothesis tests for testing both the validity of our results against surrogate data and the significance of leader-follower interactions observed in Figure 2A:

“For the chosen parameter configurations, we tested both the significance of our estimate of transfer entropy with respect to surrogate data and that of leader-follower relations observed in Figure 2A. […] All leader-follower interactions were correctly identified by this analysis and none of the significant tests is among the four cases mentioned above (Table 3, columns 5 and 6).”